# Arboreal Epiphytes in the Soil-Atmosphere Interface: How Often Are the Biggest "Buckets" in the Canopy Empty?

**Hailey Hargis [1], Sybil G. Gotsch [2], Philipp Porada [3] , Georgianne W. Moore [4], Briana Ferguson [2] and John T. Van Stan II [1,\*]**

1 Geology & Geography, Georgia Southern University, Statesboro, GA 30460, USA
2 Biology, Franklin & Marshall College, Lancaster, PA 17603, USA
3 Plant Ecology & Nature Conservation, University of Potsdam, 14476 Potsdam, Germany
4 Ecosystem Science & Management, Texas A&M University, College Station, TX 77843, USA
\* Correspondence: jvanstan@georgiasouthern.edu; Tel.: +1-912-478-8040

**Abstract:** Arboreal epiphytes (plants residing in forest canopies) are present across all major climate zones and play important roles in forest biogeochemistry. The substantial water storage capacity per unit area of the epiphyte "bucket" is a key attribute underlying their capability to influence forest hydrological processes and their related mass and energy flows. It is commonly assumed that the epiphyte bucket remains saturated, or near-saturated, most of the time; thus, epiphytes (particularly vascular epiphytes) can store little precipitation, limiting their impact on the forest canopy water budget. We present evidence that contradicts this common assumption from (i) an examination of past research; (ii) new datasets on vascular epiphyte and epi-soil water relations at a tropical montane cloud forest (Monteverde, Costa Rica); and (iii) a global evaluation of non-vascular epiphyte saturation state using a process-based vegetation model, LiBry. All analyses found that the external and internal water storage capacity of epiphyte communities is highly dynamic and frequently available to intercept precipitation. Globally, non-vascular epiphytes spend <20% of their time near saturation and regionally, including the humid tropics, model results found that non-vascular epiphytes spend ~1/3 of their time in the dry state (0–10% of water storage capacity). Even data from Costa Rican cloud forest sites found the epiphyte community was saturated only 1/3 of the time and that internal leaf water storage was temporally dynamic enough to aid in precipitation interception. Analysis of the epi-soils associated with epiphytes further revealed the extent to which the epiphyte bucket emptied—as even the canopy soils were often <50% saturated (29–53% of all days observed). Results clearly show that the epiphyte bucket is more dynamic than currently assumed, meriting further research on epiphyte roles in precipitation interception, redistribution to the surface and chemical composition of "net" precipitation waters reaching the surface.

**Keywords:** precipitation; interception; bromeliad; vascular epiphyte; non-vascular epiphyte; lichens; bryophytes; water storage capacity

---

## 1. Introduction: How Big Is the Epiphyte Bucket?

Hydrologists have long represented landscape elements along the rainfall-to-runoff pathway as "buckets" [1], or water storage elements, with various "holes" where water may escape by evaporation [2] or drainage to another landscape element, be it the litter [3], soils [4], or beyond. During storms, the first bucket that rainfall encounters in vegetated ecosystems is the plant canopy via interception [5]. Interception is the portion of rainfall that is stored by canopy elements and evaporated.

Annually, interception can return up to 45% of rainfall to the atmosphere for local forests [6] and account for as much as 80% of evapotranspiration [7]; thus, its spatiotemporal patterns influence regional moisture recycling [8] and global temperatures [9]. Canopy evaporation rates depend on water storage amounts [10] and, as such, the emptying and filling of storage buckets in the canopy can represent the dominant processes controlling total rainfall interception [11].

There are three major buckets that contribute to rainfall interception in forest canopies (and the understory): leaves, bark (often referred to as stem storage) and epiphytic vegetation [12]. Leaves, especially for forests with leafless periods, represent the smallest water storage component (Table 1). Rainwater storage capacities for the bark on branches and stems can be many times greater than estimated for leaves (Table 1) and may be an important water source for epiphytes [13]. By comparison, however, the greatest measured rainwater storage capacities are from epiphytic vegetation (Table 1). It has been reported that the global distribution of epiphytes [14] can increase rainwater storage by up to 38 times compared to the host's bare canopy [15].

**Table 1.** Range of plot-scale surface water storage capacity observations for each major canopy element. Global range of total canopy surface water storage capacity is estimated via modelling.

| Element | Range of Surface Water Storage Capacities (mm) |
|---|---|
| Leaves | 0.04–2.2 [16,17] |
| Bark | 0.2–5.9 [16,18] |
| Epiphytes | 0.4–16.6 [15,19,20] |
| Canopy roots | Unknown |
| Canopy soils | Unknown |
| Total canopy | 0.04–19 [15] |

Epiphytic vegetation consists of a wide range of forms, from true plants, like bromeliads, orchids and woody plants (some of which are facultatively epiphytic), to lichens and mosses [21]. Epiphyte buckets can include a variety of unique anatomical features besides leaf and bark surfaces. To survive in the forest canopy where they are physically disconnected from soil- and groundwater resources, and may not receive rainfall for weeks, epiphytes have developed various mechanisms to tolerate extended periods without precipitation [22–24]. Despite the abundance and diversity of desiccation tolerance strategies inherent to epiphytes, a common assumption is made that the epiphyte bucket is rarely emptied of its water, and only a small portion of their substantial water storage capacity is ever available for subsequent refilling [25–31]. Because of this, Zotz [14] states that "there seems to be a consensus that vascular epiphytes usually play a rather minor role in forest hydrology, in contrast to non-vascular epiphytes, particularly bryophytes." The attention given to bryophytes is generally due to their (and lichens') substantial water storage per unit area [12], distribution throughout forest habitats in the overstory, understory and litter layer [32,33], and rapid desiccation [34]. But even bryophytes' role in rainfall interception has been described as "rather limited despite their considerable maximum water storage capacity [as] only a fraction of the potential storage is actually available" under frequently rainy conditions [26]. Although authors of past work on epiphyte interception have been careful to qualify their results as site-specific and some report that epiphytes' rates of water loss can vary as widely as their water storage capacities [29], the assumption that the epiphyte bucket is rarely available to intercept precipitation has become common. We seek to overturn this common assumption by presenting evidence from three sources: (i) examination of published findings; (ii) incorporation of new data; and (iii) global evaluation of epiphyte saturation state for non-vascular epiphytes using the LiBry model [15].

*Anatomy of the Epiphyte Bucket*

The epiphyte bucket, arguably, contains a greater variety of elements than their host tree (Figure 1). Like their host tree, epiphytic plant communities contain leaves and can contain bark and flowers.

Unique epiphyte anatomical features, however, include leaf water pools (i.e., phytotelmata, Figure 1a,b), roots and root-like structures, lignotubers, pseudobulbs, and canopy soils. Epiphyte leaves and canopy area: the tank leaves of epiphytic vegetation, alone, have been estimated to store up to 0.34 mm of rainwater in Costa Rica [26]; however, phytotelmata can result from multiple leaf configurations in epiphytes [35] and this tank leaf water storage capacity does not include droplets stored externally (as is visible in Figure 1b). Different leaf surface structures may also differ in their precipitation interception efficiency or water storage capacity, for example: does the outer leaf surface of bromeliad tank-leaves interact differently with passing rain droplets than the foliage leaves? In addition, leaves that form bracts (i.e., structures at the base of reproductive structures) often, though not always, differ in texture, shape, size or other morphological property (typically related to rainwater storage) from the foliage leaves. Generally, the water storage capacity of foliage leaves has a wide range (Table 1) but can reach 3.3 mm for epiphyte leaves with trichomes [30]. The internal cell structure of the leaf will further alter the total leaf water storage capacity. Due to high risk of desiccation, epiphyte leaves often exhibit succulence or generally higher leaf water storage than their canopy hosts. Figure 2 provides example leaf cross-sections for vascular epiphytes of contrasting internal cell structures. For example, leaves with substantial hydrenchymal cells (i.e., specialized water storage cells that are devoid of organelles and have elastic cell walls) can drive internal leaf water storage—a clear comparison can be made between the ~1 mm of water storage capacity for the hydrenchymal rich leaves of *Clusia palmana* (Figure 2c) versus the 0.3 mm capacity of the *Oreopanax anomalus* leaves which have a thin hydrenchymal layer (Figure 2a).

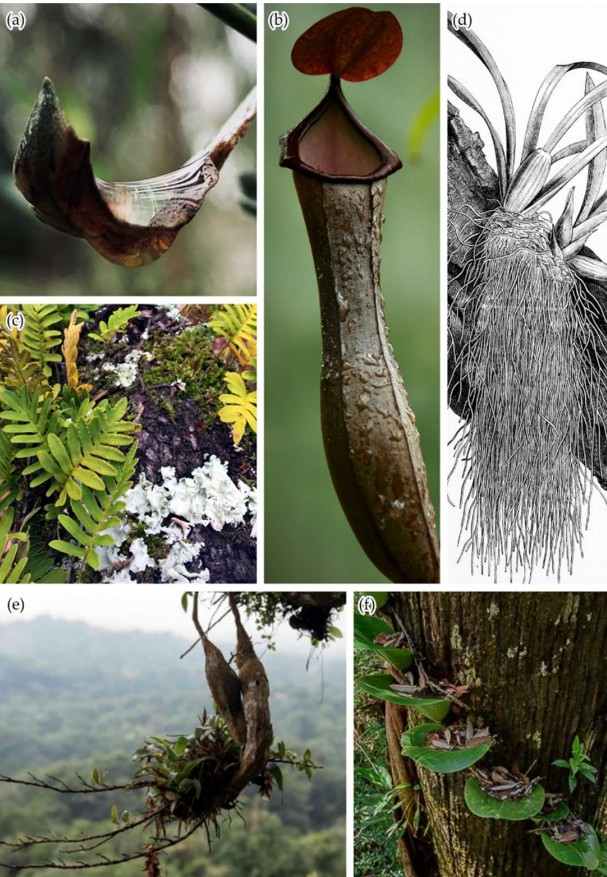

**Figure 1.** Anatomy of the arboreal epiphyte "bucket" includes: (**a**) water pools called phytotelmata [36], often held in (**b**) tank-like leaves that also externally hold water droplets [37]; (**c**) leaves and leaf-like thallus organs of lichens and bryophytes [38]; (**d**) root and rootlike organs [39] which can include (**e**) external storage structures such as lignotubers [40] and pseudobulbs (not pictured); and (**f**) trapped detritus [41].

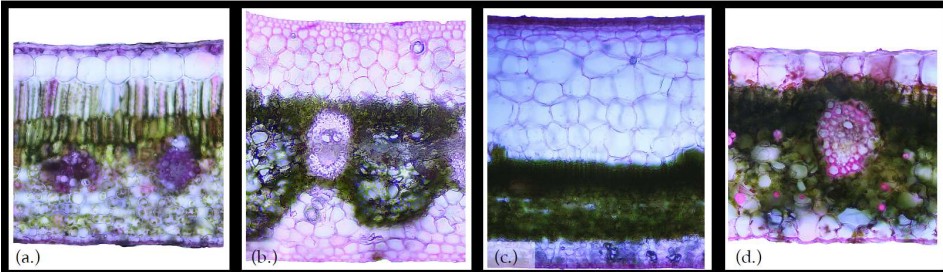

**Figure 2.** Cross sections of common vascular epiphyte leaves highlight variation in the water storage cell structure. (**a**) *Oreopanax anomalus*, a woody epiphyte, has large epidermal cells on the upper side of the leaf that likely function in water storage, and hydrenchymal cells on the underside of the leaf. The bromeliad, (**b**) *Werauhia werckleana*, has distinctive hydrenchymal layers on both sides of the leaf that comprise most of the cross-sectional area. (**c**) *Clusia palmana*, a woody hemiepiphyte, has a hydrenchymal layer on the upper side of the leaf with cells that are much larger than cells in the rest of the leaf, as well as smaller hydrenchymal cells on the underside. (**d**) *Stenospermation sessile*, an herbaceous epiphyte, does not have a distinct hydrenchymal layer; however, the oversized epidermal cells likely function in water storage and there are hydrenchymal cells interspersed throughout.

Many forest canopies, particularly in the tropics, contain substantial root biomass (Figure 1d) from vascular epiphytic plants (i.e., *Asplenium* spp.), the host tree itself [42] and neighboring plants with apogeotropic roots [43]. Adventitious roots and related ectomycorrhizal fungal networks from the host tree have even been observed to grow into the epiphyte mats [42,44], which form 30 cm to >50 cm thick layers of dead and decomposing organic matter layers [19] (discussion on the storage of detritus to come). Root-like structures, like rhizines from lichens, rhizoids from bryophytes or rhizomes from ferns (like *Pleopeltis polypodiodes*), may also be abundant in forest canopies [45]. Root biomass estimates in canopies are few and limited to the tropics, but an oft-cited estimate is 72 g m$^{-2}$ of ground surface [46]. Roots are functionally absorptive, increasing water storage capacity relative to other components, and they are found in places where rainwater is routed—inner-branch surfaces and branch junctions—increasing their likelihood of frequent saturation [46]. Rainwater uptake rates differ across epiphytes, but an example of the absorptive capability can be found in the aerial roots of various orchids which contain a spongy root epidermis layer, called velamen radicum (note: this is common among terrestrial plants, too [47]). This velamen radicum allows rapid uptake of rainwater and dissolved nutrients [48,49]—in fact, velamen radicum can begin taking up rainwater in seconds and fully saturate after 1 min [50].

A conceptual epiphyte bucket would likely be riddled with pronounced bulges, representing bulbous and tuber-like organs that store nutrients and resident animals. Epiphytic orchids, in particular, can enlarge stem sections to produce pseudobulbs [51], while other epiphyte families (e.g., Ericaceae and Rubiaceae) modify their shoots into lignotubers [52] (Figure 1e). Both structures can add substantial surface area for interception. These stem structures (and rhizomes) can be hollow, a condition typically investigated for their provision of living space to mutualistic animals, especially insects [53]; however, these void spaces could increase water storage. Since the surfaces of pseudobulbs and lignotubers are typically rougher than leaf surfaces and they may be internally porous (no study has yet estimated their water storage capacity), it may be reasonable to assume water storage capacities similar to bark estimates (Table 1). Additionally, orchid flowers can persist for months and bloom multiple times a year, adding surface area for rainwater storage.

Lastly, the epiphyte bucket is an efficient interceptor (and, as mentioned earlier, originator) of detritus falling from the canopy (Figure 1f). Leaves, stems, animal carcasses, and excrement have all been reported within epiphytic vegetation [12]. The invertebrate biomass in individual epiphytes can equal that of an entire tree crown in rainforests: ~3800 g ha$^{-1}$ [54], but the rainwater storage amounts associated with live and dead insect carcasses has not been measured. Leaf litter's detainment in epiphyte-laden canopies can be short (70% of leaf litter deposited in the canopy are lost within 2 weeks), limiting its

contribution to epiphyte nutrition, but detritus is ever-present in the canopy [55] and canopy soils often develop from it [56]. No water storage estimates currently exist specifically for epiphyte-detained detritus or related canopy soils; however, estimates are available for detritus in general, 1–20 mm [57], and for organic soils, 1–4 mm cm$^{-1}$ of soil depth [58]. Considering the anatomy of the epiphyte bucket, the cumulative water storage capacities of all these elements (from multiple community members) easily satisfies the surprisingly large, previously reported values reaching 16 mm (Table 1).

## 2. Open Buckets: Non-Vascular Epiphytes, Phytotelmata, and Canopy Soils

All forest types, and most other types of vegetated canopy, are inhabited by non-vascular epiphytes, or vegetation that lack anatomical structures or physiological functions to help maintain their internal water content. This condition, called poikilohydry, represents a vast array of lichens and bryophytes that freely uptake and release water in response to atmospheric conditions. For vascular epiphytes that rely on tank structures, the resulting phytotelmata are also free to interact with surrounding atmospheric conditions. Another component of the epiphyte bucket that allows its water content to passively interact with the physical atmospheric conditions are canopy soils. Thus, this section focuses on models that examine water storage, filling and emptying dynamics for lichens and bryophytes (LiBry [15]), the phytotelmata of pitcher plants [59], and data showing saturation dynamics of canopy soils (and, in the following section, the leaves of vascular epiphytes) at various elevations of the Monteverde Cloud Forest (Costa Rica); which demonstrate that these "open" buckets may often be available for rainfall interception.

### 2.1. Non-Vascular Epiphytes

Non-vascular epiphytes include some of the most water absorbent terrestrial organisms on Earth, reaching 300–3000% their dry weight when saturated [60]. The LiBry model simulates the dynamic growth and biomass of non-vascular epiphytes, such as lichens and bryophytes, based on climate data and other information about the environment, such as the structure of the canopy and the disturbance regime [15,61]. Thereby, the model estimates the size of the non-vascular epiphyte bucket from the amount of simulated biomass and the morphological properties of the biomass, meaning its height and porosity. The LiBry model also quantifies the dynamic water saturation of non-vascular epiphytes. Drying occurs in the model at the rate of potential evaporation, which is calculated from climatic conditions (radiation, temperature, relative humidity and wind). It is assumed that the organisms have no substantial means to prevent water loss, so their drying is comparable to that of a wet soil surface. Water uptake depends both on the available water from rainfall and on the saturation state of the non-vascular epiphyte bucket. It is assumed in the model that the fraction of rainfall which may enter the epiphytes decreases with their increasing saturation.

To test the assumption that the non-vascular epiphyte bucket is frequently close to saturation, the simulated saturation of non-vascular epiphytes was recorded during a standard model run (see Porada et al. [15,61] for details) and a frequency distribution of the saturation states was created (Figure 3). The distribution is clearly bimodal, where non-vascular epiphytes spend around 50% of their time in the dry state (35%) or in a state near saturation (15%) on the global average. Moreover, the distribution of saturation state depends on the type of ecosystem (Figure 3). In humid tropical regions, non-vascular epiphytes spend markedly more time at higher water saturation than in desert regions (note that the desert category also includes ecosystems with trees). In cooler regions, such as tundra or temperate and boreal forests, non-vascular epiphytes spend slightly more time at lower saturation values than at higher ones, compared to other ecosystem classes. While the classes differ in the time spent at lowest saturation (desert: 50%, tundra: 25%), the time spent at highest saturation is surprisingly similar between ecosystem classes (10–15%). These model results suggest that non-vascular epiphytes are not, as commonly assumed, in a state close to saturation for most of their time. On the contrary, even in the humid tropics, around a third of the time is spent in the dry state, and only 15% near saturation (Figure 3).

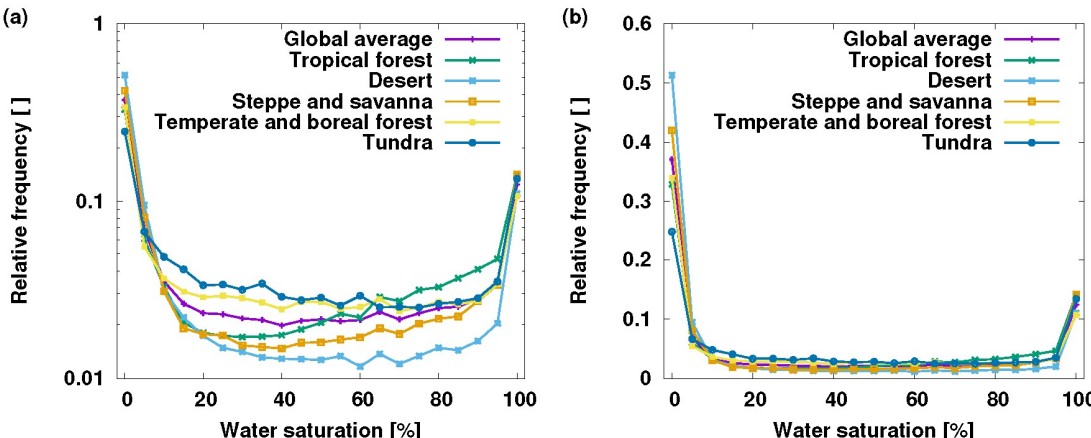

**Figure 3.** Saturation state of non-vascular epiphytes simulated by the lichens and bryophytes (LiBry) model—(**a**) frequency on a logarithmic axis to better identify differences between biomes, (**b**) the same data with a linear axis. Global average is shown with values for five ecosystem classes, based on the world's biomes [62]. Tropical forest also includes subtropical forests, and desert includes the Mediterranean biome. The resolution of the x-axis is 5%, thus the highest saturation state corresponds to the range 95–100%.

Figure 4 shows the global spatial pattern of the fraction of time, which non-vascular epiphytes spend near saturation (>95% saturation). In most regions of the world, non-vascular epiphytes spend <20% of their time near saturation. However, some regions, such as central Europe, the Sahel zone, the East of South America and parts of South-East Asia and India, exhibit higher values, between 30–70%. These high values can be largely explained by the data on leaf and stem area index (LAI and SAI) used as an input by the LiBry model. SAI and, for some evergreen forests, LAI, determine the amount of available space in the canopy which the simulated non-vascular epiphytes can use for growth. In some regions, particularly in Europe, values of LAI and SAI are relatively low compared to other regions with a similar rainfall amount (Figure S1 in Supplemental Materials). This may result from combining forest areas characterized by high SAI and potentially large epiphytic canopy storage capacity with agricultural areas which do not allow for growth of epiphytes to an average, large-scale low value. Therefore, the biomass and, consequently, the water storage capacity of the simulated epiphytes at the scale of a model grid cell is relatively low (Figure S2 in Supplemental Materials), which, combined with the relatively high rainfall (Figure S2), leads to frequent saturation.

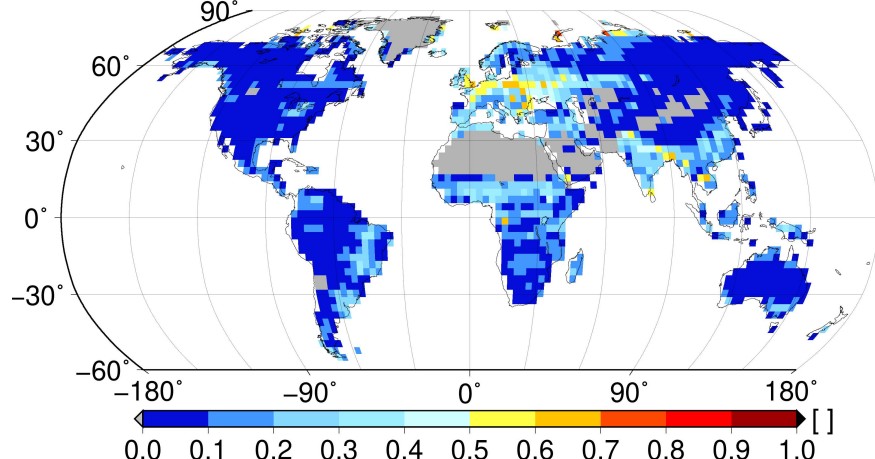

**Figure 4.** Fraction of time which non-vascular epiphytes spend near saturation (95–100% of saturation) simulated by the LiBry model. Grey areas denote regions, where no non-vascular epiphytes occur according to the model.

This result illustrates the importance of scale and model resolution for the simulated estimates. Simulated biomass is calculated from the balance of climate-driven photosynthesis and respiration, rather than from field measurements, and thus depends upon ecosystem-specific disturbance intervals that may not represent local conditions. In northern temperate and boreal forests, for instance, disturbance may be underestimated and, therefore, biomass overestimated, since logging is not considered in the model. Although simulations were consistent with the available field data, there were limited field observations available for model validation. For a detailed comparison of simulated biomass and storage capacity to observational data, and a discussion of the uncertainties in model estimates, see Porada et al. [15]. To examine this further, the LiBry model was forced with climate data from a local site in Costa Rica (Soltis Center for Research and Education, San Isidro de Peñas Blancas) and, additionally, with the corresponding climate data from the global data set (the model grid cell which includes Costa Rica). The global climate data are based on the Water and Global Change (WATCH) data set [63]. This site has reported stand-scale canopy interception of 12.1% of incident rainfall for several weeks during the mid-summer (June to July) [64]. Figure 5 shows the estimated distribution of non-vascular epiphyte saturation state for both simulations. While both distributions are bimodal, the global climate data lead to a higher estimated time at low saturation compared to the local data (32% vs. 15%). Vice versa, the local climate data are associated with a substantially higher fraction of time near saturation compared to the global data (35% vs. 13%). The local estimate of time fraction near saturation agrees with past research at the site [65]. This significant discrepancy between local and global estimates by the LiBry model may be explained by the following differences between the two climate data sets: rainfall is 60% higher and solar radiation is 30% lower in the local data set compared to the global one, which means less potential evaporation. Wind speed is 60% lower in the local data set, which further reduces potential evaporation. Hence, the local climate data represent significantly more humid conditions than the global data, which explains the higher simulated amount of time spent by the epiphytes near saturation. It should be noted that the local climate data only include one year (2014). Therefore, the global data were averaged over the period 1958–2001 to an 'average' year, since 2014 was not available in the global data set. This may have further increased differences between the two climate data sets.

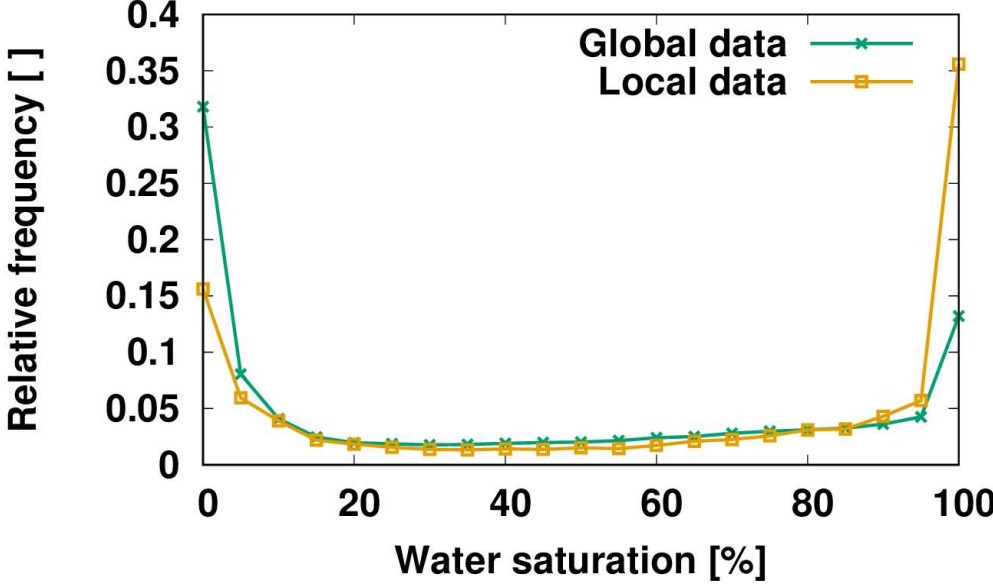

**Figure 5.** Saturation state of non-vascular epiphytes simulated by the LiBry model for two different climate data sets. Local data were recorded in 2014 at the Soltis weather station in Costa Rica. Global data are based on the average of the years 1958–2001 of the WATCH data set (see Porada et al. [15]) for the grid cell (2.8125° × 2.8125°) which includes Costa Rica.

For high values of water saturation, rainfall explains most of the differences between simulations driven either by local or global climate data. This causes more frequent full water saturation in the local simulation. At low values of water saturation, however, wind and solar radiation are more important factors than rainfall for the less frequent drying in the local simulation compared to the global one (see Figure S3). To summarize, local conditions in the climate and, therefore, in the non-vascular epiphyte storage capacity and water saturation, may substantially differ from large-scale estimates. In tropical montane regions, like the grid cell that includes Costa Rica, landscape heterogeneity can cause substantial microclimatological variation over short distances—even within the same elevation [66,67]. It is unclear, however, which data are more suitable to draw conclusions on the general, average saturation state of the (non-vascular) epiphyte bucket. The local data may not be representative of the average climatic conditions over large regions, while the global data may not be suitable to describe certain habitats, which may have a significant influence on the average saturation state at the large scale. Despite these uncertainties, a consistent result of the LiBry simulations is the frequently occurring low and intermediate saturation of non-vascular epiphytes, which contradicts the common assumption that these organisms are usually close to saturation.

## 2.2. Phytotelmata

Epiphytic vascular plants that have developed leaf structures to capture precipitation (and associated nutrients) have also captured significant scientific attention. The overwhelming majority of research on these plants, typically bromeliads with tank-like leaves that can accumulate voluminous phytotelmata, investigates ecophysiological processes that occur when these leaf water pools are present. The few studies on these plants when empty and the frequency of this "dry" state, find that their phytotelmata may often dry out completely [59,68,69]. Persistence of phytotelmata depends primarily on plant size [59] and, at least for the plants studied, this relationship between plant size and drying time was unaffected by leaf orientations as extreme as 60° from vertical [69]. Even the phytotelmata of large epiphytic bromeliads have been shown to dry out completely within a few days under typical meteorological conditions for tropical forest canopies [59]. For smaller bromeliads, Zotz and Thomas [59] reported that they were likely without water nearly one in every three days or approximately 110 days per year, based on hydrometeorological data from their rainy study site (~2600 mm y$^{-1}$). Drying rates for phytotelmata can be quickened by 30–60% for all plant sizes when local environmental changes remove overhead canopy cover [69]. In addition, strong winds were observed to cause spillage of the phytotelma [69], which would likely increase the length of time that plants are empty and, therefore, available to intercept rainwater. As these observations are limited to a few epiphytic bromeliads, and phytotelmata-containing structures are diverse in epiphytes, more observations on the water levels of phytotelmata under field conditions are needed.

## 2.3. Canopy Soils

Various detrital sources are captured by epiphytes and create canopy soils (Figure 1f). Water holding capacity of detritusphere O-horizon soils (like Of, Om, and Oh, collectively called Ol) can be relatively high [58], because of the high absorbency of newly-formed organic matter, i.e., humus [70]. Seasonal variability in both moisture and detrital sources will influence the size of the canopy soil reservoir [19]. Canopy soil, meteorological and canopy hydrologic data are rarely collected together, but one ongoing study in the tropical montane cloud forest (TMCF) region in Monteverde, Costa Rica, is collecting such data. Researchers have established six canopy sites along an elevation gradient spanning from 1100 m on the Pacific side of the mountain range, which is a premontane rain forest, to 1600 m on the Atlantic side of the range, which is an exceedingly wet and relatively aseasonal cloud forest site [71]. We present, as an example, an analysis of the frequency of canopy soil dryness at multiple elevations for Monteverde's forest canopy (Figure 6)—a site that represents conditions where the canopy saturation state is expected to be consistently saturated. The period of observation represents a misty and windy transition period (December–January), a dry season

(February–April) and the beginning of the wet season (May). The uppermost site (1550 m asl) is in cloud forest, the mid-elevation (1400 m asl) site is just below the average current cloud base and the lowest site (1100 m) is in premontane rain forest. Further site details have been published previously [71], but each of these sites host canopy soils. Available canopy soil water storage varied with elevation, but a large portion of the maximum soil moisture content was often available to precipitation (and occult deposition) throughout the study period and many small rain events did not fill canopy soils to capacity (Figure 6). Available canopy soil water storage was greatest for canopy soils at the lowest site (1100 m), where the canopy soil was <50% saturated only 29% of the 182 days. The site at 1400 m, experiencing some occult deposition, remained below the 50% saturation threshold 40% of the total observation period. Interestingly, the site with most frequent occult deposition (i.e., the true "cloud forest" site at 1550 m asl) was <50% saturated on 53% of all days observed. This could be associated with greater overall volumes of canopy soil in cloud forests [19]. These values exceed frequencies for non-vascular epiphytes (in the previous analysis), yet non-vascular epiphytes tend to have higher surface areas that dry more readily [24]; whereas, soils may retain soils longer due to lower surface-to-volume ratio. Even so, available canopy soil water storage is arguably high for a system that is considered saturated all or most of the time and, again, counter that the common assumption that components of the epiphyte bucket are usually close to saturation.

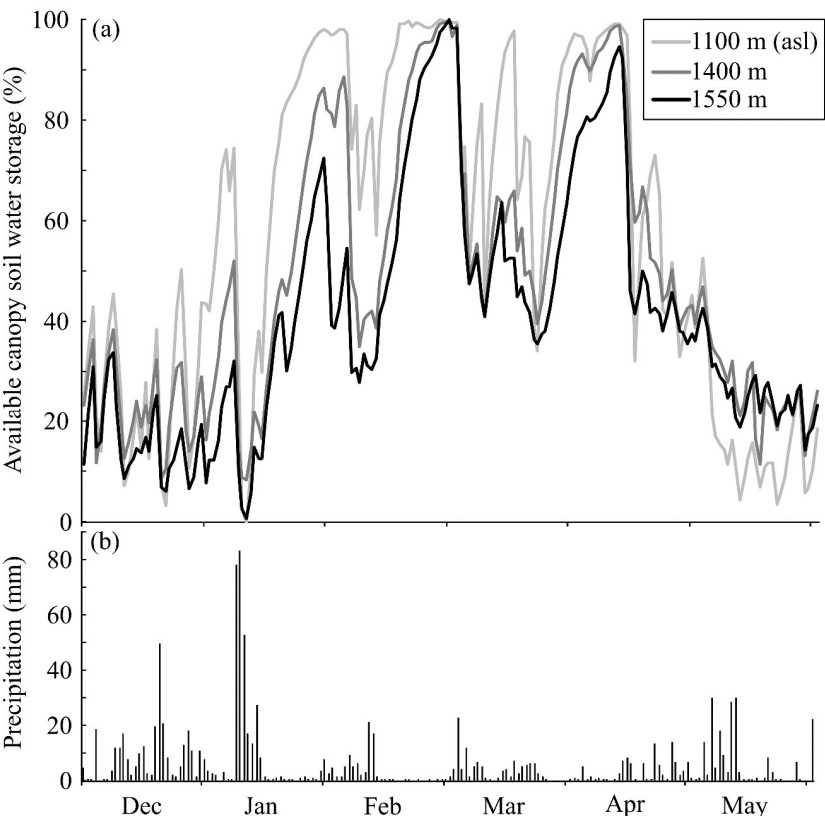

**Figure 6.** (**a**) Available canopy soil water storage of canopy organic matter, December 2016 to May 2017 at three sites in a tropical montane region in Costa Rica. Volumetric soil moisture content was measured with a soil moisture probe (METER Group, EC-5). Since the soil moisture probes are still deployed, calibration of soils was not possible. Since some probes logged negative values in the dry period in this study due to this lack of calibration, data were scaled across sites by assuming that the lowest value in the driest part of the dry season was 0 (mm$^3$/mm$^3$) in all sites. The max. value was set as 100% saturation, then the inverse of that was plotted to show the available canopy soil water storage. (**b**) Precipitation at the uppermost site is plotted (while total precipitation is lower at lower elevations, the patterns of rainfall are similar in all sites).

### 3. Vascular Epiphytes: Do Their Ecophysiological Mechanisms Keep the Bucket Full?

For the vascular epiphyte bucket, it is important to note that current hydrologic theory limits interception water storage solely to the plants' externally held water (it being available to physical evaporative drivers) [72]. Functionally, however, interception is the portion of precipitation that does not reach the surface. Through this lens, the physiological capture, storage and transpiration of precipitation by vascular epiphytes may be considered a component of interception, although transpiration is frequently suppressed in wet canopies due to humid/moist conditions [65]. Vascular epiphytes employ a diversity of ecophysiological mechanisms to take up rainwater rapidly [30,50,56,73]. These mechanisms may enhance surface water retention and delay leaf drying—e.g., some species can retain 4 times more water per unit area and dry out 12 times slower [74]—but these mechanisms may not keep the bucket full for two reasons: (i) many of the water uptake mechanisms employed by vascular epiphytes involve highly absorptive external components, like trichomes or velamen radicum, that dry out rapidly after a storm [30,50]; and (ii) rapid water uptake mechanisms are not always coupled with water conservation or storage mechanisms. Rather, many vascular epiphytes, particularly ferns, couple rapid water uptake mechanisms with desiccation tolerance mechanisms [73,75,76]. In fact, trade-offs in vascular epiphytes have been found between allocation to traits that promote foliar water uptake and traits that promote water storage [22], indicating that vascular epiphytes can either hold onto water in leaves or readily uptake water via leaf surfaces—probably not both.

Vascular epiphytes exhibit a wide array of structures on and in leaves to capture and store water. Trichomes, hairs and leaf scales intercept water and in some cases these have been shown to funnel rainwater toward central plates and into the plant [77,78]. Due to the structure of trichomes, they retain substantial rainwater externally and, although the water that was taken up can be stored (in hydrenchyma), the external water is evaporated rapidly. The aforementioned absorbent velamen radicum that aid root uptake of rainwater for a wide variety of plants [47] can dry within 15 min when thin, but even thicker roots dry within a several hours [50]. Even when specialized trichomes are not in occurrence on leaf surfaces, during periods of high humidity, water will condense on leaf surfaces where it can either be shed, absorbed or evaporated. Leaf hydrophobicity and microclimate will largely determine the partitioning between these three processes. In a recent review, Dawson and Goldsmith estimate that, across all ecoregions in the world, leaves are wet an average of ~100 d yr$^{-1}$ and in tropical and subtropical forests leaf wetness occurs ~174 d yr$^{-1}$ [79]. If there is space in the leaf to receive externally held water, the high frequency in the capacity for different species to perform foliar water update indicates that this water can and is often absorbed by the leaf [22,80] (S.G. Gotsch et al., unpublished data).

To determine the importance of the internal leaf "bucket" in canopy water cycling, a quantification of leaf water storage across species and habitats is necessary as well as the quantification of changes in leaf water stores with dehydration. In an ongoing study, Gotsch et al. [22,71] have quantified functional traits in sites along an elevation gradient in the TMCF in Monteverde, Costa Rica. Since the cloud base is projected to rise in this region, concern is mounting regarding the vulnerability of the TMCF ecosystem and canopy plants in particular [22,81–83]. While there is substantial variation across species and sites, on average, vascular epiphyte leaves have an internal leaf water storage capacity of 0.5 mm (Figure 7). The average internal leaf water storage ranges from 0.7 mm in the lowest elevation site, which is currently below the cloud base, to 0.4 mm in the cloud forest. In all sites, there are several species that have an internal leaf water storage of 0.9 mm or greater. These species belong to the genera *Clusia*, *Hillia*, *Columnea*, *Peperomia*, *Notopleura*, and *Pleurothallis*—all of which are common in the study sites (Figure 2c).

If this leaf water storage is stable over time then its importance to canopy water cycling would be diminished, though there is evidence that this storage component is actually quite dynamic. In two drought experiments on vascular epiphytes in the TMCF, leaf thickness exhibited marked declines in most species as water availability decreased, and this pattern was reversed with a rehydration period following the drought [84,85]. The reduction in leaf thickness ranged from <10% in non-succulent

plants to 80% for more succulent plants [84,85]. Water storage occurs primarily in hydrenchymal cells in the leaf as seen in the leaf cross-sections in Figure 2. These cells tend to have high elasticity in general (low bulk elastic modulus) and have greater elasticity than surrounding tissues which allows them to function as a supply of water to the photosynthetically active portions of the leaf during water shortage then refill when water is in abundance [86]. Since water loss and carbon gain are linked via stomatal openings, this water can also be pulled from the leaf via the transpiration stream when the stomata are open. Measures of cuticular conductance of these epiphytes range from 3.0 mmol m$^{-2}$ sec$^{-1}$ in wetter sites to 6.0 mmol m$^{-2}$ sec$^{-1}$ in drier sites (S.G. Gotsch, unpublished data). While these values are lower than those recorded for terrestrial plants in tropical rainforests, they make clear that, even when the stomata are closed, water loss occurs via leaf surfaces of vascular epiphytes.

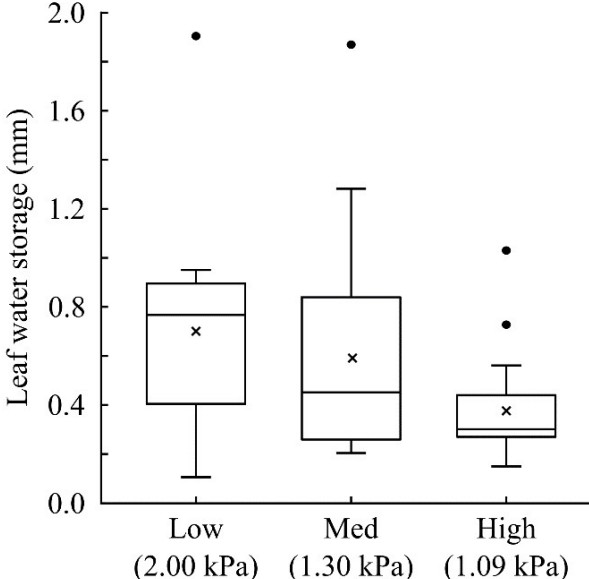

**Figure 7.** Leaf water storage of vascular epiphyte communities in three cloud-affected forests in a tropical montane region in Monteverde, Costa Rica. Maximum vapor pressure deficit for each site is provided parenthetically. The site labelled "Low" is a pre-montane rainforest, the middle elevation site ("Med") is near the cloud base, and the "High" site is a 1600 m cloud forest on the Atlantic slope of the continental divide. In each site, 15–30 species were measured (*n* > 5 individuals per species). The mean is indicated by an "x", the median by a line, the 25–75% quartile by the box, the maximum and minimum of the non-outlier range by the whiskers, and any outliers are shown as dots.

Many vascular epiphytes have ecophysiological mechanisms that permit near-complete water loss from their leaf tissues, and potentially represent another type of "open" bucket for rainfall interception. These so-called "resurrection plants" can live in a dehydrated state (i.e., as a very empty bucket) between storm events, then rapidly rehydrate during storms (within 48 h) and dehydrate again within days [87,88]. Over 330 species have been identified as resurrection plants to date, and these include many epiphytic ferns. Epiphytic ferns like *Rumohra adiantiformis*, *Pyrrosia lingua*, *Cheilanthes myriophylla*, *Asplenium ceterach* and *Pleopeltis polypodioides* can survive leaf water contents as low as 41%, 27%, 10%, 4% and even 3%, respectively, and can rapidly dry near to these levels given proper meteorological conditions [73,75]. These water contents are striking in comparison to the dehydration limitations of plants outside of this paraphyletic group, which die after losing only 5–20% of their leaf water contents [76]. Dehydration rates in resurrection plants cannot occur too quickly, however, or the plant cannot survive [87]; e.g., the most rapid dehydration of any resurrection plant measured so far has been 4–5 h for an aquatic plant [89,90]. Still, the arboreal epiphytic fern, *P. polypodioides*, along the Georgia (USA) coast, for example, often dries within a few days after a storm (J.T. Van Stan II, unpublished data). Since the median inter-storm dry period for the region in 2015–2016 was 111 h

and the interquartile range is 40–168 h [91], it is likely that *P. polypodioides* is often in a dehydrated state prior to the start of any proceeding storm event. While resurrection plants are common across a number of ecosystems, *P. polypodioides* can be particularly abundant in humid subtropical forests of the southeastern US, where their biomass reaches nearly 0.5 kg m$^{-2}$ of branch surface (12–46 mg cm$^{-2}$ from twenty-five ~40 cm$^2$ samples collected from *Quercus virginiana* hosts; see Figure S4 for photographs) and submersion tests (similar to Van Stan et al. [30]) found specific water storage capacities for this species reaching 2.1 mm. In these ecosystems especially, the external and internal water relations of epiphytic resurrection plants represent a rapid cycling and important canopy "bucket".

## 4. Conclusions

The ecohydrological literature generally agrees that, when present, the epiphyte "bucket" can represent a large portion of precipitation water storage capacity in forest canopies. The common assumption that this bucket is nearly always full, and thus not available to intercept precipitation, however, is not supported by the results of this work. Rather, our examination of published findings, analysis of new datasets, and global evaluation of the saturation state for non-vascular epiphytes using the LiBry model, all indicate that much of the water storage capacity of arboreal epiphyte communities is available to intercept precipitation across scales and ecosystems. We want to point out, however, that substantially more field observations are needed to improve validation of water storage capacity simulated by the LiBry model, particularly for extra-tropical regions. Since the amount of field observations will likely remain too low for direct empirical upscaling, future work may also focus on collecting consistent local data sets of climate variables, epiphyte biomass and storage capacity. In this way, mechanistic models of epiphyte water saturation could be better validated and the estimated large-scale values would be more reliable. Globally, LiBry results show non-vascular epiphytes spend <20% of their time near saturation. Across regions and ecosystems, even in the humid tropics, model results find that around one-third of the time is spent in the dry state for non-vascular epiphytes. Even local data from sites with substantial atmospheric moisture receipt (Costa Rican cloud forest sites) showed the (i) epiphyte community was saturated only one-third of the time, (ii) their internal leaf water storage was highly dynamic (being replenished by precipitation interception), and (iii) their related canopy soils were often <50% saturated (29–53% of all days observed). These results clearly show that the epiphyte bucket is more dynamic than currently believed and that these dynamics may play significant roles in the return of precipitation to the atmosphere which, in turn, can influence the surface energy balance. Further investigation of the water relations of arboreal epiphyte communities also addresses recent calls by the hydrologic community to move beyond the water balance and begin inventorying "compartmentalized stores and the connections between them" [92]. Finally, arboreal epiphytic vegetation can be found in forest types across all major climate zones: e.g., tropical [93], subtropical [30], arid [94,95], temperate [96], and boreal [97]. Future work on the partitioning of precipitation by vegetation should, therefore, consider the epiphyte community's role.

**Supplementary Materials:** The following are available online at http://www.mdpi.com/2076-3263/9/8/342/s1: Figure S1: Annual average values of (a) leaf area index and (b) stem area index used in the LiBry model. Figure S2: Global maps showing patterns of normalized (top) canopy water storage capacity and (bottom) rainfall amount. Figure S3: Saturation state of non-vascular epiphytes simulated by the LiBry model for five different climate data sets. Figure S4: Photographs showing an example mat of abundant Pleopeltis polypodioides (resurrection fern).

**Author Contributions:** Conceptualization, H.H. and J.T.V.S.II; methodology, all authors; software, P.P.; validation, all authors; vascular epiphyte analyses, S.G.G., G.W.M. and B.F.; data curation, G.W.M.; writing—original draft preparation, H.H. and J.T.V.S.II; writing—review and editing, all authors; visualization, all authors; supervision, J.T.V.S.II and S.G.G.

**Funding:** Funding for research at the Soltis Center site was supported by the U.S. Department of Energy, Office of Science, Biological and Environmental Research (DE-SC0010654). Support for ongoing research in the TMCF of Monteverde Costa Rica comes from the from National Science Foundation (IOS Award #1556289 to S.G.G). P.P. gladly appreciates funding by the Deutsche Forschungsgemeinschaft (DFG, German Research Foundation)—408092731.

**Conflicts of Interest:** The authors declare no conflict of interest.

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
