# Peer review of "Arboreal Epiphytes in the Soil-Atmosphere Interface: How Often Are the Biggest “Buckets” in the Canopy Empty?"

_geosciences, doi:10.3390/geosciences9080342_

Round 1
Reviewer 1 Report
This article provides very interesting results about the water storage of nonvascular epiphytes and the spatial and temporal variation. It give new insights about the conventional assumptions. Overall, i'm supportive of publication of this work. Some minor revisions listed below are needed before final acceptance.
Please modify the vertical axis of fig.3 because the interval is not equal and difficult to get direct proportions of the the time frequency.
For some regions in Europe, the time duration of nearly saturated water condition is higher. The authors attributed the reasons to the low value of LAI ans SAI. So, how is the variation of LAI and SAI globally, the authors could add the result to make it more robust and concrete.
The authors are encouraged to make a comparison of the global and local climate data to find the most important variable influencing the results presented in Fig.5.
Author Response
Please modify the vertical axis of fig.3 because the interval is not equal and difficult to get direct proportions of the time frequency.
RESPONSE: In the revised manuscript, we added another panel in Figure 3, which has a linear vertical axis, so the absolute values can be better read. We have also extended the caption: "... a) frequency on a logarithmic axis to better identify differences between biomes, b) the same data with a linear axis."
For some regions in Europe, the time duration of nearly saturated water condition is higher. The authors attributed the reasons to the low value of LAI and SAI. So, how is the variation of LAI and SAI globally, the authors could add the result to make it more robust and concrete.
RESPONSE: In the revised manuscript, we have added global maps of leaf and stem area index (Figure S1) to the supplementary material.
The authors are encouraged to make a comparison of the global and local climate data to find the most important variable influencing the results presented in Fig.5.
RESPONSE: The revised manuscript includes a new figure in the supplementary material, which shows the impacts of individual climate variables on the frequency distribution of water saturation. Moreover, we will add the following to the main text (at line 255): "For high values of water saturation, rainfall explains most of the differences between simulations driven either by local or global climate data. This causes more frequent full water saturation in the local simulation. At low values of water saturation, however, wind and solar radiation are more important factors than rainfall for the less frequent drying in the local simulation compared to the global one (see Figure S3)."
Reviewer 2 Report
While the work is generally of high quality, and it treats an important topic, there are serious shortcomings that must be addressed before the work should be published.
My primary concern is the general lack of knowledge and basis for this work in temperate through boreal forests. Their cited literature amply illustrates this, where all but a few citations on volumes or capacities are in the tropics. There is scant work on canopy epiphyte occurrence and abundance outside of the tropics. The authors cite the relevant literature, such as it is, but we are basically ignorant here. Almost all studies that exist appear to be biased to old growth or lightly disturbed forests, so capture extreme high values, but are likely unrepresentative, given the cutting history in temperate forests, particularly in the northern hemisphere. While proportion capacity may be estimated, one needs capacity to estimate volumes, and it appears that simply is not possible given what we know to date.
The authors can't address this lack of basic measurements in a single paper or analysis. It appears they attempt to address modeling uncertainties through models across a parameter space, but it is unclear that they have addressed this primary determinant of storage explicitly. I spent a great deal of time reviewing the methods details in the current manuscript, and in the primary citation of Porada et al., 2013 in Biogeosciences, and in the citations traced back therein, and it isn't clear this lack of knowledge about realized capacity is treated, and its central importance in fluxes isn't addressed. True, this paper is about relative frequency of saturation, but this question is unimportant outside the tropics/subtropics if capacity is near nil for most locations, which is a plausible, even likely, condition.
I'm aware that the lacking information may be in the supplementary materials, which they cite in their paper, but which I can't access.
So how should they address this lack of basic information? By acknowledging it specifically in their analysis, and specifically describing modeling analysis across the plausible ranges of ground and canopy bryophytes for these regions. A thorough job would probably require more than one or two additional papers.
For this paper, it would suffice to 1) expand their discussion to highlight this lack of knowledge for extra-tropical regions, and 2) modify their maps and output to reflect this lack of knowledge. They should at least include an uncertainty map, generated from the range of parameters they use, showing volume uncertainty in estimates. Better yet, they could simply truncate the maps and estimates for extra-tropical regions, in the same way they do using the landcover/LAI maps as a filter, which one assumes drives the saturation frequency for the Sahara and much of Greenland to zero.
Formatting and attention to detail in the references section other major shortcoming of the paper. There are abundant errors in capitalization, incomplete citations, and inconsistencies in reference format and abbreviation. These would have to be fixed before publication, but makes one wonder about attention to detail in the other components of this work.
I would recommend rectifying both of these problems as mandatory before publication.
Author Response
My primary concern is the general lack of knowledge and basis for this work in temperate through boreal forests. Their cited literature amply illustrates this, where all but a few citations on volumes or capacities are in the tropics. There is scant work on canopy epiphyte occurrence and abundance outside of the tropics. The authors cite the relevant literature, such as it is, but we are basically ignorant here. Almost all studies that exist appear to be biased to old growth or lightly disturbed forests, so capture extreme high values, but are likely unrepresentative, given the cutting history in temperate forests, particularly in the northern hemisphere. While proportion capacity may be estimated, one needs capacity to estimate volumes, and it appears that simply is not possible given what we know to date.
The authors can't address this lack of basic measurements in a single paper or analysis. It appears they attempt to address modeling uncertainties through models across a parameter space, but it is unclear that they have addressed this primary determinant of storage explicitly. I spent a great deal of time reviewing the methods details in the current manuscript, and in the primary citation of Porada et al., 2013 in Biogeosciences, and in the citations traced back therein, and it isn't clear this lack of knowledge about realized capacity is treated, and its central importance in fluxes isn't addressed. True, this paper is about relative frequency of saturation, but this question is unimportant outside the tropics/subtropics if capacity is near nil for most locations, which is a plausible, even likely, condition.
RESPONSE: We agree with the reviewer that the little available field data on water storage capacity in temperate and boreal forests may indeed be biased towards old-growth forests and high values. However, we want to point out that the LiBry model does not use any field observations of storage capacity as input to calculate corresponding large-scale values. Instead, simulated storage capacity is derived from biomass, which is calculated based on the balance of photosynthesis and respiration (based on climate), and also depends on ecosystem-specific disturbance intervals. This means that uncertainty in the field observations of water storage capacity caused by a low amount of measurements is not automatically reflected in the model estimates. The model covers all regions in the northern hemisphere, and therefore predicts a broad range of storage capacities (see Porada et al. 2018, cited in manuscript). Land use is also considered in the model (which explains the relatively low leaf and stem area index in Europe). However, the simulated disturbance intervals are based on natural processes and logging is not considered. Hence, we added the following to the revised manuscript (line 228): “Simulated biomass is calculated from the balance of climate-driven photosynthesis and respiration, rather than from field measurements, and thus depends upon ecosystem-specific disturbance intervals that may not represent local conditions. In northern temperate and boreal forests, for instance, disturbance may be underestimated and, therefore, biomass overestimated, since logging is not considered in the model. Although simulations were consistent with the available field data, there were limited field observations available for model validation.”
So how should they address this lack of basic information? By acknowledging it specifically in their analysis, and specifically describing modeling analysis across the plausible ranges of ground and canopy bryophytes for these regions. A thorough job would probably require more than one or two additional papers.
RESPONSE: In the paper by Porada et al (2018) the simulated biomass and water storage capacity of lichens and bryophytes are evaluated using the available field observations from the literature. The generally low amount of field data, particularly in extra-tropical regions, is discussed there. We point this out in the revised manuscript by appending the following to the new text above: “For a detailed comparison of simulated biomass and storage capacity to observational data, and a discussion of the uncertainties in model estimates, see Porada et al.”
For this paper, it would suffice to 1) expand their discussion to highlight this lack of knowledge for extra-tropical regions, and 2) modify their maps and output to reflect this lack of knowledge. They should at least include an uncertainty map, generated from the range of parameters they use, showing volume uncertainty in estimates. Better yet, they could simply truncate the maps and estimates for extra-tropical regions, in the same way they do using the landcover/LAI maps as a filter, which one assumes drives the saturation frequency for the Sahara and much of Greenland to zero.
RESPONSE: In the revised manuscript, we highlight the low amount of field observational data on water storage capacity, particularly in extra-tropical regions (see above). However, we think that this does not represent a fundamental problem for our modeling approach since, as explained above, the model does not rely on field observations of biomass and water storage capacity for calculating large-scale values. Instead, biomass and storage capacity are derived from climatic data and ecosystem properties, such as leaf and stem area index or disturbance intervals. Indeed, one motivation for our alternative, climate-based large-scale modeling approach was the lack of field data on storage capacity, which makes empirical upscaling difficult. We carried out a sensitivity analysis in Porada et al (2018), which showed that simulated storage capacity is not very sensitive to variations in the modeling approach. In particular, the uncertainty in the modeled storage capacity applies equally to all regions around the world, since the model parameters are mostly derived from global studies. Hence, we do not think that the simulated maps of water saturation should be truncated. In contrast to the simulated values, the uncertainty in the observed storage capacity may indeed be larger in the northern regions than in the tropics, due to a lower number of studies in extra-tropical regions. However, in Porada et al (2018) we show that the amount of suitable data is so low in any region of the world, that it is impractical to calculate standard deviation values for ecosystems. This leads to uncertainties considering model validation. To clarify this, we added the following to the conclusions of the revised manuscript (at line 443): “We want to point out, however, that substantially more field observations are needed to improve validation of water storage capacity simulated by the LiBry model, particularly for extra-tropical regions. Since the amount of field observations will likely remain too low for direct empirical upscaling, future work may also focus on collecting consistent local data sets of climate variables, epiphyte biomass and storage capacity. In this way, mechanistic models of epiphyte water saturation could be better validated and the estimated large-scale values would be more reliable.”
Formatting and attention to detail in the references section other major shortcoming of the paper. There are abundant errors in capitalization, incomplete citations, and inconsistencies in reference format and abbreviation.
RESPONSE: The reference list of the revised manuscript has been generated from the endnote MDPI template and manually checked for consistency.